# A Single Injection of ADRCs Does Not Prevent AAA Formation in Rats in a Randomized Blinded Design

**DOI:** 10.3390/ijms25147591

**Published:** 2024-07-10

**Authors:** Egle Kavaliunaite, Pratibha Dhumale, Charlotte Harken Jensen, Søren P. Sheikh, Jes S. Lindholt, Jane Stubbe

**Affiliations:** 1Cardiovascular and Renal Research Unit, Institute for Molecular Medicine, University of Southern Denmark, 5230 Odense M, Denmark; 2Department of Cardiac, Thoracic and Vascular Surgery, Odense University Hospital (OUH), 5000 Odense C, Denmark; jes.sanddal.lindholt@rsyd.dk; 3Department of Clinical Research, University of Southern Denmark (SDU), 5230 Odense M, Denmark; pratibhadhumale09@gmail.com (P.D.);; 4Department of Clinical Biochemistry, Odense University Hospital (OUH), 5000 Odense C, Denmark; 5Open Patient Data Explorative Network, Institute of Clinical Research, Odense University Hospital (OUH), 5000 Odense C, Denmark

**Keywords:** stem cells, abdominal aortic aneurysm, regenerative medicine

## Abstract

There is a pressing need for alternative medical treatments for abdominal aortic aneurysms (AAAs). Mesenchymal regenerative cells derived from adipose tissue (ADRCs) have shown potential in modulating the inflammation and immune responses that drive AAA progression. We hypothesized that ADRCs could reduce inflammation and preserve vascular integrity, potentially slowing the progression of AAA. In our study, subcutaneous adipose tissue was harvested from male Sprague Dawley rats, from which ADRCs were isolated. AAA was induced in these rats using intraluminal porcine pancreatic elastase, followed by intravenous administration of either ADRCs (10^6^ cells) or saline (0.1 mL). We monitored the progression of AAA through weekly ultrasound, and the rats were sacrificed on day 28 for histological analysis. Our results showed no significant difference in the inner abdominal aortic diameter at day 28 between the control group (172% ± 73%, n = 17) and the ADRC-treated group (181% ± 75%, n = 15). Histological analyses of AAA cross-sections also revealed no significant difference in the infiltration of neutrophils or macrophages between the two groups. Furthermore, the integrity and content of elastin in the tunica media were similar between groups. These findings indicate that a single injection of ADRCs does not inhibit the development of AAA in rats in a randomized blinded study.

## 1. Introduction

An abdominal aortic aneurysm (AAA) is a permanent pathological dilation of more than 3 cm that weakens the aortic wall. AAAs tend to expand further and, if left untreated, may result in a rupture of AAA, resulting in a mortality of over 80% of the patients [1]. Until AAA expands to a diameter of 5.5 cm in men and 5 cm in women, no treatment is offered, and the strategy is to surveil the growth of AAA with ultrasound measurements until it becomes so large that surgical repair is offered, which can either be performed by endovascular or by open surgery. However, both of these surgeries carry their own risks [2]. Unfortunately, there is no medical treatment to attenuate AAA development and progression that could prevent or delay the need for surgical repair or the risk of rupture.

The underlying cause of AAA remains unknown, but the pathogenesis of AAA is thought to be a multifactorial process that includes the infiltration of inflammatory immune cells and the degradation of the extracellular matrix components mediated by proteases in particular matrix metalloproteinases (MMPs) [3]. Infiltrated inflammatory cells such as macrophages, lymphocytes, and neutrophils cause the activation of matrix metalloproteinases, particularly MMP-2 and MMP-9, which result in the degradation and fragmentation of elastin, activating vascular smooth muscle cells (VSMCs) to compensatory synthesize collagen and elastin and other ECM components, but they will eventually become exhausted and undergo apoptosis [4]. All these essential elements contribute significantly to AAA progression and the further weakening of the aortic wall [5,6,7]. The loss of the normal balance of the connective tissue production and degradation leads to the inevitable weakening of the aortic wall [3,4].

In the last decade, mesenchymal stem cells (MSCs) have been suggested as an effective treatment for myocardial infarction and other diseases with vascular injury, due to the promising therapeutic modalities [8].

Multiple preclinical studies have reported positive findings when investigating the mechanisms and efficacy of MSC intervention in AAA disease models [9,10]. These findings further support previous reports of MSCs being able to participate in vascular remodeling by suppressing elastin degradation, inhibiting MMP-2/9 and the proinflammatory cytokine tumor necrosis factor (TNF) expression, thereby preserving the VSMC in their contractile phenotype [11,12].

Although recent studies report a positive association between MSC treatment and the attenuation of AAA progression, the studies lack blinding and randomization. Blinded and randomized animal studies are sparse; however, these aspects are crucial for confidently translating treatment effects from animals to humans.

In this study, we examined the therapeutic effect of adipose-derived regenerative cells (ADRCs) in the development of AAA using a rat porcine elastase perfusion aneurysm model in a randomized blinded design. The decision to use freshly isolated ADRCs in the present study was based on the fact that they can easily be obtained and show pluripotency, with the ability to develop into various cell types including osteoblasts, adipocytes, chondrocytes, and even endothelial cells, helping to reinforce the structural integrity of the aortic wall and regeneration [13]. We hypothesize that ADRCs can attenuate the progression of AAA by its anti-inflammatory properties.

## 2. Results

Forty rats were intended to be included in the experiment. Two rats died after surgery, and 6 rats were euthanized due to chronic pain and failure to flourish on days 7 and 10. Thus, we ended up with n = 17 in the control group and n = 15 in the ADRC-treated group. No difference was observed in body weight neither at the beginning of the experiment nor at the termination of the experiment (Table 1). At the termination of the experiment after 28 days, the heart, spleen, liver, and kidneys were weighed to assess systemic side effects on vital organs. No changes regarding organ-to-body weight were observed between the two groups (Table 1).

### 2.1. ADRC Treatment Did Not Alter AAA Progression

The weekly maximal AAA diameter at the end-systole was compared between the ADRC treatment and control groups. Both groups developed AAA after surgery by day 7 and continued to grow steadily until day 14, whereafter AAA expansion only slowly progressed until day 28. However, no statistically significant differences between the groups were observed after 28 days (Figure 1, n = 17/15, *p* = 0.7) nor on day 0, day 7, day 14, and day 21. These results suggest that ADRCs are not able to affect the AAA expansion rate.

### 2.2. ADRC Treatment Did Not Affect Elastic Fiber Content

Even though AAA size did not differ between groups, we further analyzed if ADRC treatment had any effect on the aortic aneurysm’s wall integrity. Elastin composition in tunica media was evaluated at the termination of the experiment 28 days after AAA induction. Firstly, an evaluation of the remaining percentage of elastin in tunica media was assessed. There was no significant difference in the percentage of elastin remaining in tunica media between the groups (Figure 2a,b, n = 16/15, *p* = 0.64). Secondly, the elastin samples were assessed for the elastin degradation score, as described previously [14]; both groups showed extensive defragmentation and loss of elastin, and no difference was observed between the groups (Figure 2e, n = 16/14, *p* = 0.94). Thus, ADRCs do not appear to affect elastin degradation.

### 2.3. ADRC Treatment Did Not Alter the Infiltration of Neutrophils and Macrophages

The ability of ADRCs to halt the infiltration of inflammatory cells was assessed by evaluating the number of neutrophils determined by myeloperoxidase (MPO)-positive cells and macrophages by the CD68-positive stained area in aneurysmal cross-section samples in the treatment and control groups. There was no difference in the number of the neutrophils of the aneurysmal wall between the control group and the treated group (Figure 3, n = 17/15, *p* = 0.76). Also, the percentage area of infiltrating macrophages (CD68 positive staining) did not differ between groups (Figure 4, n = 15/14, *p* = 0.39). These data suggest that ADRCs do not affect the inflammation status in the aneurysm wall at day 28 post-surgery.

## 3. Discussion

Our present result did not confirm our hypothesis demonstrating that a single intravenously administered injection of ADRC could attenuate AAA progression in the PPE-induced AAA rat model. On the contrary, we observed no effect on the AAA expansion rate, changes in aneurysm wall elastin content or integrity, nor in the infiltration of neutrophils or macrophages after providing ADRCs systemically shortly after AAA induction. This is in contrast with a recent systematic review and meta-analysis by Li et al. [10], summarizing 18 studies all showing a protective effect of stem cell treatment in AAA development in pre-clinical studies. The authors concluded that stem cell treatment is promising; however, the authors also assessed the risk of publication bias, and the low animal number included in some of the studies, and acknowledged the risk of overestimating, at least parts of, the protective effect of the treatment. Although most studies in the review used cultured mesenchymal stem cells, while our study utilized ADRCs where only a subset of cells were mesenchymal stem cells, we believe that due to the high potency of ADRC, the effect should at least be comparable. Furthermore, Li et al. reviewed the quality of the studies and deemed it to be low due to the lack of randomization and blinding. Before our study, we performed power calculations to ensure that the number of animals included provided the power to draw meaningful results based on clinical settings. We further performed all experiments by investigators blinded to treatment to avoid potential investigator bias. In addition, it is well established that each pre-clinical AAA model reflects different aspects of the human disease [15] and could also partly explain the discrepancy between our results and the previously reported data. It is also plausible that other research groups did not report negative findings when testing the effect of stem cells. However, to the best of our knowledge, our preclinical study is the first report analyzing stem cells as a treatment option for AAA attenuation using a randomized blinded approach that could perhaps partly explain the discrepancy when comparing our results with previous studies.

Although many studies report a positive association between stem cell treatment and the ability to halt the progression of AAA, the precise mechanism of the stem cell effect is still unknown. Many praise stem cells for their ability to differentiate into various cell types [9], though only one study successfully reported ADRC locally differentiating [16]. Many studies show decreased degradation of extracellular arterial elastin after stem cell treatment when compared to the control group [17]. An increase in elastin could have an important role in maintaining the integrity of arterial elastin, a crucial element for the prevention of AAA initiation, development, and rupture. In our study, ADRCs did not have any obvious effect on elastin content at post-surgery day 28. We cannot rule out that at an earlier time point, where inflammation is at its highest around day 7, there might have been an effect. However, based on the AAA size, it is less likely that elastin integrity would have been highly affected.

Other reasons why ADRC treatment did not affect AAA expansion could be (1) the time point of ADRCs delivery, (2) that a single injection of ADRCs was not enough to alter the progression of AAA, (3) the number of ADRCs was too low, or (4) ADRCs should have been further enriched by CD31+ selection, which is known to be responsible for enhancing the vascular regenerative potential of ADRCs in the treatment of erectile dysfunction, at least in human settings [18]. There is no consensus on when it is the optimal time to initiate treatment with ADRCs. While some studies opt for pre-treatment [19], others start treatment after AAA initiation [14,20] to best mimic human AAA disease progression where the aim is to halt already existing AAA.

It is also plausible that multiple injections with ADRCs would be necessary to have a long-lasting attenuation of AAA in humans. Only one previous study investigated weekly intravenous ADRC injections throughout their study period of four weeks and found the treatment to be effective [21]. In our model, we did not observe an immediate effect regarding AAA size to treatment in the first week or following weeks. Perhaps the number of cells used in our single injection was too low. We used 1 × 10^6^ cells per injection which was based on successful previous experience, where ADRCs were delivered to the penis of the rat to treat erectile dysfunction [22]. Thus, at least in erectile dysfunction in rats, the number of cells worked as intended and was in line with another AAA preclinical study [10]. However, other preclinical AAA studies with similar designs injected a higher number of cells, 6 × 10^4^ to 9 × 10^6^ [16,23], which could suggest that a larger number of cells is needed to be effective in AAAs. Finally, the local administration in the penis study may have allowed more time for the ADRCs to secrete paracrine factors, whereas, in our study, the systemic administration would likely cause the ADRCs to pass the affected arterial wall faster, possibly reducing the window of action.

It is worth mentioning other general limitations of using animal models for chronic diseases such as AAA. Firstly, aortic injury in animals is always caused by acutely degrading elastin [15], whereas, in humans, it could take years for an aneurysm to form [3]; thus, the balance between degradation and ECM remodeling is skewed. Secondly, our study was designed to initiate treatment immediately after the induction of AAA by elastase perfusion. As AAA is almost always asymptomatic when developing, thus taking years to be diagnosed, treating AAA immediately after development would be impossible. A small murine AAA study inducing AAA by angiotensin II by subcutaneously implanted osmotic mini pumps has examined stem cell-based treatment on already formed AAA at day 28 [24] rather than in the acute phase at day 0 or 1 post-surgery in the PPE-AAA model. Though their study was underpowered (n = 5), and mice were allocated to treatment randomly rather than based on AAA size on day 28, the observed reverted effect of AAA size by stem cell treatment should be interpreted with some caution. Their chosen window for treatment may be more clinically relevant than in our chosen time point. However, our aim was to affect the rapid expansion that occurs within the first two weeks of AAA expansion and is driven by inflammation and oxidative stress, as we hypothesized that ADRCs would dampen the inflammatory response and prevent AAA expansion.

Another limitation is a rather short follow-up period. Most studies in rodents include either 14 days follow-up or 28 days with only a few studies reporting extensive follow-up [25]. This period allows researchers to detect a response to treatment in an animal model; however, long-term response in humans remains unclear. It is also worth mentioning that most studies, including ours, design their experiments using a single one-time injection. Although our study did not find a positive association between a single injection and the attenuation of AAA, other studies have shown that a single injection was able to attenuate AAA development [11]. If implemented in clinical practice, multiple stem cell injections would likely be necessary to halt the progression of AAA for a lifetime; thus, a single injection approach would need to be reassessed. Future studies would focus on multiple injection models. Arterial injections are avoided due to the high risk of embolization, leading to severe complications. Venous injections are the standard practice and are considered safe for patients in both current and future treatments. We acknowledge that not examining ADRCs’ homing is a study limitation, as it would have shown whether ADRCs reached the AAA wall. Since the rats were only euthanized at the experiment’s end, it is uncertain if ADRCs could be detected in the wall at that time. Additionally, the timing and duration of treatment are crucial elements. A rat study using the same PPE AAA model demonstrated a significant decrease in AAA expansion 14 days post-induction with 1 × 10^6^ human umbilical cord mesenchymal stem cells administered intravenously on post-surgery day 1 [17]. This implies that the timing and quantity of ADRC delivery were expected to be effective in our experiment. Thus, the observed difference is likely ascribed to the different subpopulation of regenerative cells used. Furthermore, weekly cell delivery, as shown by Fu et al. [21] in an AngII-induced AAA model in apoe^−/−^ mice, could have been considered. Despite the limitations of the current AAA models, these models are powerful and sufficient to understand large areas of the pathophysiological mechanisms that underlie AAA and to anticipate therapies.

Another strength of our study is using ultrasound to precisely track diameter changes of AAA throughout our follow-up period. As ultrasound is the main diagnostic tool used for human AAA surveillance, utilizing this method allows for the possibility of mimicking human surveillance, and the possibility of measuring inner-to-inner diameters rather than measuring macroscopically at the endpoints of the experiment, providing more insights into AAA progression.

In conclusion, in contrast to former studies, our results of this strictly blinded experiment show that ADRCs given just after AAA induction do not affect AAA progression. Based on these findings, a single injection of ADRCs does not prevent AAA formation in rats in a randomized blinded design.

## 4. Materials and Methods

Ethical statement and experimental animal housing conditions.

All animals were cared for according to the ARRIVE guidelines [26] for the use of animals in research. The study was performed in accordance with protocols approved by the Danish Animal Experiments Inspectorate (license nr.2021-15-202100499).

The male Sprague Dawley rats were purchased from Janvier Laboratories (Le Genest-Saint-Isle, France). Upon arrival at our animal facility, rats had at least one week to acclimatize before the induction of AAA. All rats were housed in pairs of 4 and had free access to water and standard chow, under constant temperature (20 °C) and humidity (55%) with a 12 h light/dark cycle at the Biomedical Animal Facility at the University of Southern Denmark. We only used males, as females are generally protected against AAA formation [27,28].

### 4.1. Study Design

This design of the study was set up like a randomized blinded trial. The rats were allocated to either a single ADRC injection (1 × 10^6^ cells/100 μL PBS) or a single PBS injection (100 μL, control) 30 min after the end of AAA induction surgery. The allocation for the treatment was arranged by investigators not involved in the execution of the experiment. All animals underwent an inguinal fat pad resection on the same day as the AAA induction, and an external allocator assigned each animal to either the treatment group or the control. Moreover, ADRC harvesting was performed by investigators not involved in the execution of the animal experiments and provided the animal investigator with numbered filled and covered syringes, thus allowing the operators to be blinded to the allocation throughout the experimental settings.

### 4.2. Outcomes

The primary outcome was the aneurysm inner-to-inner anterior-posterior diameter of the systolic-peak infrarenal aortic aneurysm. Secondary outcomes included morphological analyses of elastin content in the AAA wall, the infiltration of neutrophils (myeloperoxidase, MPO-positive cells), and the infiltration of macrophages assessed by CD68-positive staining.

### 4.3. Sample Size Calculation

Based upon earlier studies [14], sample size calculation concluded that 24 rats (12 in each group) are needed to detect a 50% reduction in aneurysmal progression rate, which is considered clinically relevant. Earlier training for the experiment showed not all rats survived AAA induction due to technical difficulties during surgery. Thus, to ensure we reached the power of the study, 40 rats were included in the experiment.

### 4.4. Induction of AAA by Porcine Pancreatic Intraluminal Elastase (PPE) Infusion in Male Rats

Induction of AAA was performed as previously described [14]. Briefly, 0.2 mg of Temgesic (buprenorphine, Indivior, North Chesterfield, VA, USA) was mixed preoperatively in 1 g of nut paste (Nutella, Alba, Italy) for postoperative pain management. For surgery, animals were anesthetized with a subcutaneous injection mixture of fentanyl (236 μg/kg), fluanisone (7.5 mg/kg, Skanderborg Apotek, Skanderborg, Denmark), and midazolam (3.75 mg/kg, Hameln Pharma, Hamelin, Germany). Through a midline incision, the infrarenal aorta was meticulously dissected, exposed, and ligated at the proximal and distal ends. Then, a transverse arteriotomy was performed close to the distal ligation, which was followed by a catheter insertion (Tygon tubing, 0.01 × 0.03-inch, Qusina, Cusano Milanino, Italy) for direct intraluminal porcine pancreatic elastase (PPE) (Sigma-Aldrich, Søborg, Denmark) perfusion. The elastase infusion consisted of 10 units/mL, resulting in aortic expansion of 150%, and was kept for 30 min. Afterward, the catheter was removed, the abdomen flushed with saline, the arteriotomy was closed with a suture, ligatures were removed, and flow was restored. After surgery, every animal was injected with 5 mL of sterile physiological saline subcutaneously and placed back in the cages. Postoperative pain was managed for the following day with 0.2 mg of Temgesic mixed in 1 g of Nutella.

### 4.5. ADRC Isolation

On the morning of the day of AAA induction, all rats underwent inguinal fat pad resection for harvesting adipose tissue under isoflurane (Sigma-Aldrich, Søborg, Denmark) sedation.

Tissue dissection and ADRC isolation were performed essentially, as previously described [29]. Briefly, approximately 0.5–1 g of subcutaneous white adipose tissue was resected from the right hind leg of each rat and rinsed by placing the tissue in 15 mL tubes with 3 mL of DMEM. The resected fat tissue was further cut into smaller pieces of 2–4 mm using sterile scalpels, and ADRCs were isolated using the enzyme-based MACS Adipose Tissue Dissociation Kit (Miltenyi Biotech, 130-105-808, Lund Sweden) together with a gentleMACS Octo Dissociater with heaters (Miltenyi Biotech, Lund Sweden) according to the manufacturer’s recommendations. Enzymes were inactivated by the addition of 5 mL of DMEM/1%Pencilin/Streptomycin (PS)/10%Fetal bovine serum (FBS) and the mixture was filtered through a 100 µm cell strainer. ADRCs were pelleted by centrifugation at 500× *g* for 10 min and washed with 10 mL of DMEM/1%PS/10%FBS. ADRCs were pelleted again (500× *g*, 5 min), resuspended in DMEM and passed through a 40 µm filter (Falcon) before being counted using a Nucleocounter^®^ NC-200 (ChemoMetec, Alleroed, Denmark). ADRCs were washed once more in PBS before finally being resuspended to ADRC 1 × 10^7^ cells/mLPBS) and kept at room temperature until injection (less than 5 min). These cells, though isolated by a slightly different method but based on similar principles, have previously been cultured and shown to be relative homogenous CD45−/CD31−/CD34−/CD44+/CD90+ cells with the ability to differentiate into various cell types [22].

The injection was performed via the tail vein 30 min after the completion of the AAA surgery. The control animals received a tail vein injection of 100 μL of PBS while the treatment group received 1 × 10^6^ cells in 100 µL of PBS.

### 4.6. Ultrasound Measurements of Aneurysm Expansion

All ultrasound scans and measurements were performed with the same ultrasound machine (LogiQ e with L10-22-RS transducer, GE HealthCare, Wauwatosa, WI, USA). An anterior to posterior inner-to-inner diameter was measured each time.

The baseline ultrasound measurement was performed before the induction of AAA on the day of surgery. Afterward, the ultrasound measures were repeated once weekly; 7, 14, 21, and 28 days after the induction of the AAA. Rats were anesthetized by using 4% isoflurane inhalation anesthesia (Sigma-Aldrich, Søborg, Denmark). Scanning and analysis of the ultrasound scans were performed by the same investigator blinded to the treatment group.

### 4.7. Termination

All animals were terminated after 28 days post AAA induction by exsanguination. Internal organs, namely the liver, spleen, lungs, heart, and kidneys, were collected. Lungs were macroscopically investigated for lung injuries, such as infarcts or ischemia. The liver, spleen, lungs, heart, and kidneys were weighed.

The AAA was dissected and cut transversally into two pieces; the proximal part was fixed in a 10% normal formalin buffer (Hounisen Laboratory A/S, Skanderborg, Denmark) for 24 h at 4 °C, then placed in phosphate-buffered saline (PBS) (Thermo Fisher, Slangerup, Denmark) with 0.05% azide (Sigma-Adrich, Søborg, Denmark) and kept at 4 °C until embedding in paraffin, which was later used for further histological analyses.

### 4.8. Histology and Immunohistochemical Analysis

AAA cross sections were sectioned at 5 μm. For the Elastin detection, Miller’s elastin staining was performed following the manufacturer’s instructions (Atom Scientific, Hyde, UK). MPO and CD68 immunohistochemical staining was performed as previously described [20].

Micrographs were captured using an Olympus Bx51 microscope with an attached Olympus 162 DP26 camera (Olympus Danmark, Søborg, Denmark), and micrographs were stitched together using Microsoft Image Composite Editor 2.0. For the assessment of histological tissue Image J software (ImageJ 1.53a Wayne Rasband, National Institutes of Health, Bethesda, MD, USA) was used. The percentage of elastin in each sample was evaluated using the color threshold tool. Elastin content was also assessed by scoring aneurysmal wall disruption. For this evaluation, each sample picture was divided into 8 fields, and each field was given a score from 1–4, 4 being severe wall disruption and 1 being minimal wall disruption [20].

The number of MPO-positive cells was normalized to the total area of the AAA sample (cells/mm^2^). The % CD68 positive labeling is divided by the total area of the AAA sample. All analyses were performed by an investigator blinded to treatment.

### 4.9. Statistical Methods

The data were analyzed using GraphPad Prism (9.0 Mac OS X, Boston, MA, USA) software, and all data sets passed the normality test by passing the Shapiro–Wilk test, and results are expressed as mean ± standard deviation (SD) with a statistical significance set at *p* < 0.05. A two-way ANOVA with repeated measurements was used to analyze the aneurysm diameter between groups. Two-tailed unpaired Student’s *t*-test was used for histological evaluations, body weights, and tissue weights between groups.

## Figures and Tables

**Figure 1 ijms-25-07591-f001:**
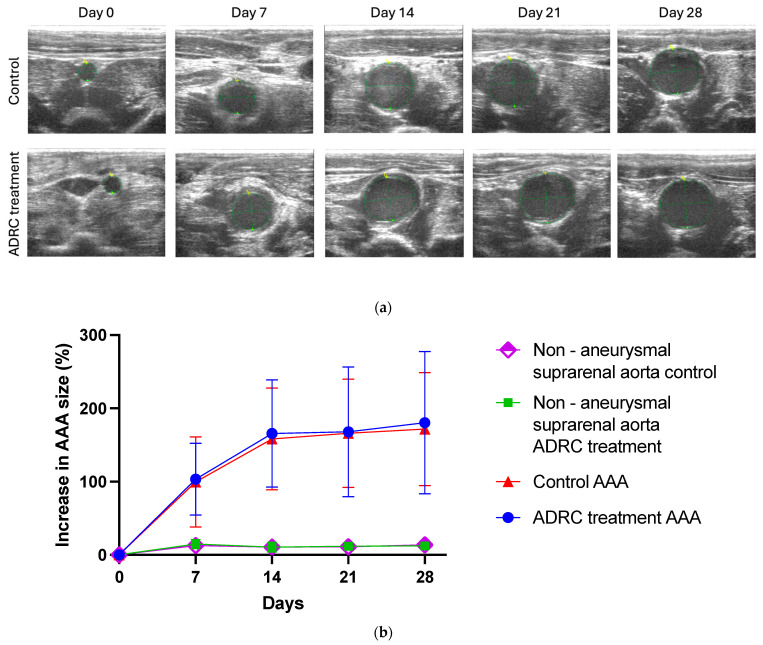
The treatment effect of ADRCs on abdominal aortic aneurysm progression. (**a**) Representative examples of ultrasound scans of AAA measurements in the control group and ADRC treatment group prior to surgery and 7 days, 14 days, 21 days, and 28 days post-surgery. C: Circumference (green circle), mm; A: area of green circle mm^2^; d1: inner to inner diameter (yellow 1+: mm); d2: Diameter (mm). (**b**) Percentage increase of AAA after initiating AAA till 28 days after treatment in the control group (n = 17) and the ADRC treatment group (n = 15), as well as percentage increase of suprarenal aorta in the control group (n = 17) and the suprarenal aorta in the ADRC treatment group (n = 15). Values are presented as mean ± standard deviation.

**Figure 2 ijms-25-07591-f002:**
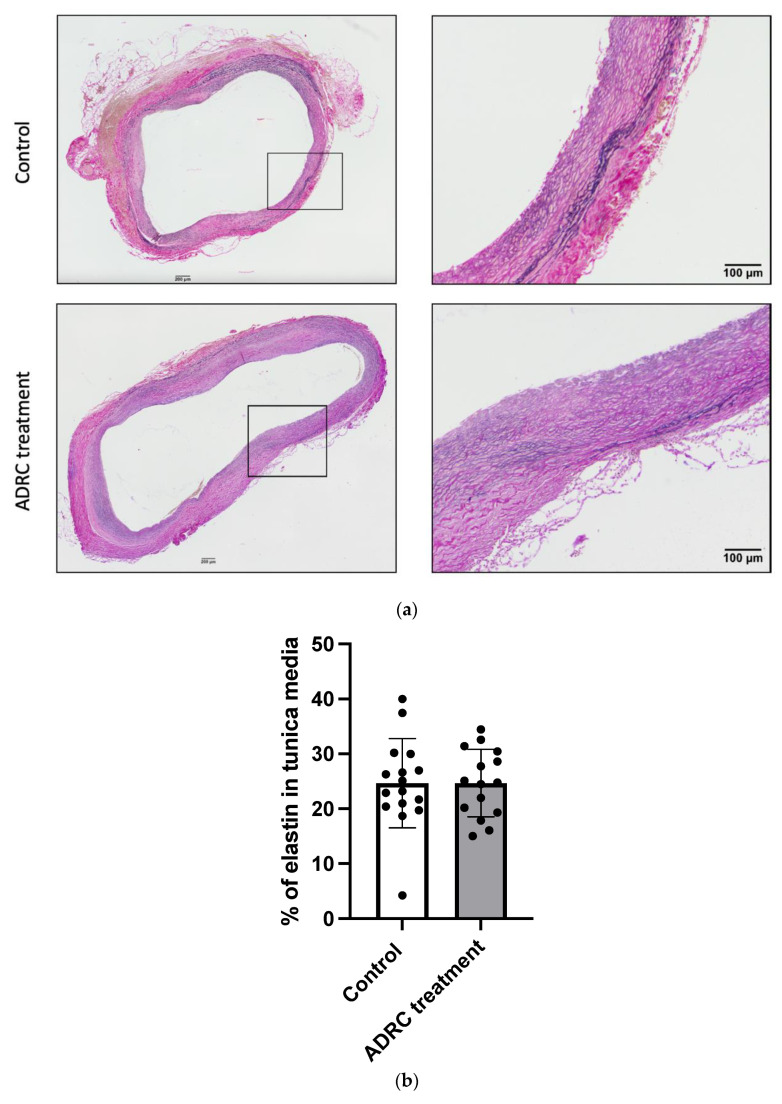
ADRC effect on elastin in the aneurysm wall. (**a**) Representative micrograph of aneurysmal cross-sections in the control group and ADRC-treated group. (**b**) The percentage of elastin in tunica media on day 28 in the control group and the treatment group (n = 17/15). (**c**) Example image of a cross-sectional abdominal aortic aneurysm divided into eight areas. (**d**) Below, micrographs representing the score of elastin degradation (1 = preserved elastin architecture; 4 = total disruption of complete concentric elastin lamellae). (**e**) The average mean score of elastin degradation in tunica media in the control group and the treatment group (n = 17/15). Values are presented as mean ± standard deviation. Each dot represents the quantification of the individual rats.

**Figure 3 ijms-25-07591-f003:**
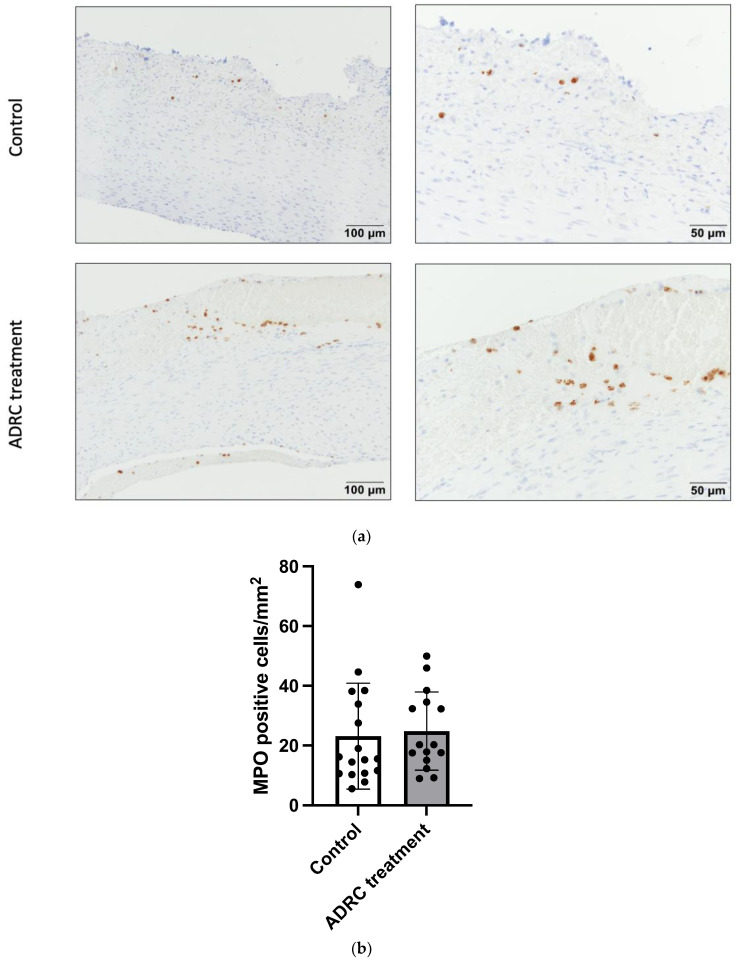
Infiltration of neutrophil cells in AAA wall. (**a**) Representative micrographs of MPO staining from aneurysmal tissue in the control and the ADRC treatment groups; left micrographs 20× magnification, right 40×. (**b**) Number of MPO-positive cells per mm^2^ in the two groups. Each dot represents the quantification of the individual rats. Values are presented as mean ± standard deviation (n = 17/15).

**Figure 4 ijms-25-07591-f004:**
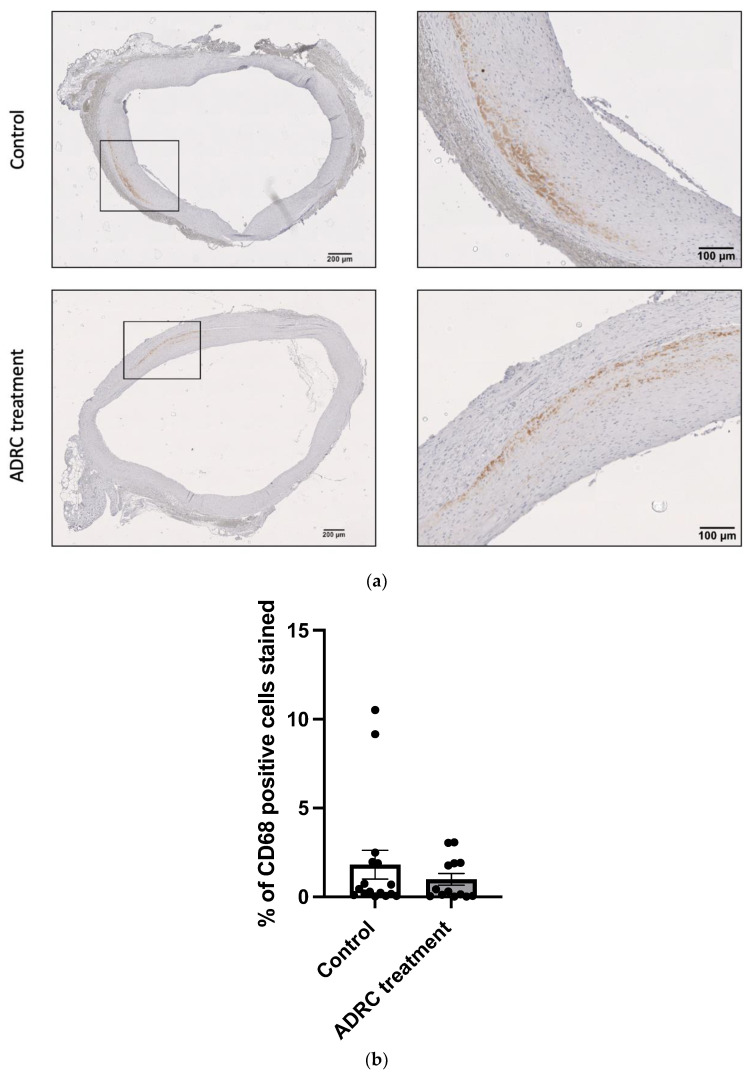
Infiltrating macrophages into the aneurysmal wall. (**a**) A representative micrograph of CD68 staining from aneurysmal tissue in the control and the ADRC-treated group. (**b**) Percentage of CD68 cells’ positive stained area (n = 16/13). Each dot represents the quantification of the individual rats. Values are presented as mean ± standard deviation.

**Table 1 ijms-25-07591-t001:** Organ-to-body weight ratio at termination day 28.

Organ-to-Body Weight Ratio (mg/g)	Mean of the Control Group (n = 17)	Mean of the ADRC Treatment Group (n = 15)	*p*-Value
Liver	33.7 ± 2.4	33.74 ± 2.4	0.97
Kidney	5.81 ± 0.7	5.95 ± 0.6	0.55
Heart	3.28 ± 0.3	3.27 ± 0.3	0.89
Spleen	2.65 ± 0.4	2.59 ± 0.4	0.63
Pre-operative weight	427.6 ± 35.1	427.1 ± 37.0	0.97
Weight at termination	535.4 ± 44.5	533.6 ± 45.1	0.91

## Data Availability

Data are available upon request.

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
