# Peer review of "A Single Injection of ADRCs Does Not Prevent AAA Formation in Rats in a Randomized Blinded Design"

_ijms, 2024, doi:10.3390/ijms25147591_

Round 1
Reviewer 1 Report
Comments and Suggestions for Authors
The authors investigated therapeutic effects of adipose-derived mesenchymal regenerative cells (ADRCs) against abdominal aortic aneurysms (AAA) in elastase-induced AAA rats in a randomized-blinded design. Analysis of aortic diameters, elastin degradation and infiltration of inflammatory cells showed no significant difference between AAA and control. The authors concluded that ADRCs did not inhibit AAA progression. However, scientific concern is that the data is insufficient and over-interpreted.
Questions
1. As the authors mentioned in the discussion, cell number and frequency of ADRCs are important issues in cell therapy. Fu XM, et al suggested that the different cell dose may affect the strength of anti-inflammatory effects of mesenchymal stem cells (PMID: 23875706). Unless the authors investigate them, it cannot be concluded that ADRCs do not prevent AAA progression.
2. Although isolation ADRCs from rat adipose tissue was performed based on the methods that were described in their previous paper, ADRCs in the previous study were obtained from a mouse. The definition of ADRCs used in this study is unclear. The authors need to prove that mouse ADRCs are mesenchymal stem cells apart from the previous study because the animal species are different.
3. The authors evaluated infiltration of macrophages and neutrophils. However, micrographs were not very clear even in the high-resolution images. In addition, immunohistochemistry experiments resulted in weak signals. Better micrographs should be presented.
4. Did the authors investigate expression levels of MMP-2, and -9? The author should investigate them.
5. Did the authors examine homing and differentiation of ADRCs at the site of AAA?
6. The title needs to be changed to “A single injection of ADRCs do not prevent AAA formation in rats in a randomized-blinded design”.
Comments on the Quality of English LanguageThere are several mistakes in representing numbers and units.
Author Response
Major issues are raised:
We thank the reviewer for the comments, these are our responses.
Reviewer 1:
- As the authors mentioned in the discussion, cell number and frequency of ADRCs are important issues in cell therapy. Fu XM, et al suggested that the different cell dose may affect the strength of anti-inflammatory effects of mesenchymal stem cells (PMID: 23875706). Unless the authors investigate them, it cannot be concluded that ADRCs do not prevent AAA progression.
RESPONSE: Yes, we agree that the timing, number of cells delivered, and way of delivery are crucial to identify the window where ADRCs would be most effective and provide the anti-inflammatory and regenerative properties we aimed for.
The number of cells and frequency of administration are crucial elements for successful treatment, we have also addressed these questions in the discussion, lines 245-254. We agree that it is plausible there is a dose-respondent effect, however, a meta-analysis by Li et al. (PMID: 35209940) summarized that dose of 1 x10^6 was a widely used dose with positive effects in other studies. The lack of cell tracking is a significant limitation of our study. However, multiple studies describe a single venous injection through the tail vein as the preferred route of administration, providing an established drug administration method. Li et al (PMID: 35209940) further summarized that 21 out of 28 reviewed studies testing stem cell treatment in AAA development administered stem cells intravenously.
Furthermore we will as suggested by you modulate our conclusion to ”A single injection of ADRCs do not prevent AAA formation in rats in a randomized-blinded design”.
Lines 32, 33 and 275, 276 changed accordingly to reflect this.
- Although isolation ADRCs from rat adipose tissue was performed based on the methods that were described in their previous paper, ADRCs in the previous study were obtained from a mouse. The definition of ADRCs used in this study is unclear. The authors need to prove that mouse ADRCs are mesenchymal stem cells apart from the previous study because the animal species are differentv.
RESPONSE: We agree we cannot fully guarantee, that the rat ADRCs are like the mouse ADRCs. However, studies have shown that while minor variations exist in proliferation rates and specific secretome profiles between rat and mouse MSCs, these differences do not significantly impact their overall utility in research and therapeutic contexts. In both species MSCs maintain essential characteristics such as multipotency, immunomodulatory abilities, and similar surface marker expression profiles, which are critical for their application in regenerative medicine​ (PMID: 23592838, PMID: 18544014).
Furthermore, we have previously isolated and cultured ADRCs from rats, though using a different isolation protocol but based on similar principles in Quade et al (PMID: 35163613) though isolated by a slightly different method but based on the same principle with collagen digestion: Here cultured ADRCs from rats were characterized to be relative homogenous and consisted of CD45−/CD31−/CD34−/CD44+/CD90+ cells and exhibited an adipogenic-, chondrogenic- and osteogenic differentiation potential as well as the ability to form capillary-like structures, when stimulated.
We have included in the method section: “These cells have previously been cultured though using a slightly different isolation method and shown to be relative homogenous CD45−/CD31−/CD34−/CD44+/CD90+ cells with the ability to differentiate into various cell types (Quaade et al. 26).” Lines 349-351 changed.
Another article with similar to our protocol for ADSCs (PMID: 25722428), reported the cells to be, by a large fraction (about 80%) of the CD11b+ cells macrophages, as indicated by the F4/80 marker. Within this group, a significant majority expressed the M2 macrophage phenotype marker CD301, suggesting that these cells are involved in tissue repair and anti-inflammatory responses.
In addition, we have been provided limited time to edit the paper, it does not allow us to perform a further characterization of the cells used.
- The authors evaluated infiltration of macrophages and neutrophils. However, micrographs were not very clear even in the high-resolution images. In addition, immunohistochemistry experiments resulted in weak signals. Better micrographs should be presented.
RESPONSE: Sorry for the poor resolution. We have updated the micrographs. In addition, we have updated the enlargements for the MPO staining as the full aneurysm cross-section did not clearly display any MPO-positive cells.
- Did the authors investigate expression levels of MMP-2, and -9? The author should investigate them.
RESPONSE: It is a fair suggestion to further examine matrix degradation and tissue remodeling as seen by others e.g. Fu et al. (PMID: 23875706). However, our analysis of the primary outcome was effect on AAA growth by diameter (d1) did not suggest any differences between controls and stem cell treated rats. Furthermore, we did not detect any significant differences in Miller elastin content nor severity score of elastin breakage in the aneurysm wall between groups. These findings suggest that no major differences in protease activity like MMP2 or MMP9 between ADRC treated rats and control rats. Based on the limited time given to review the paper, we are unable to conduct these additional analyses.
- Did the authors examine homing and differentiation of ADRCs at the site of AAA?
RESPONSE: We agree that lack of homing and examination of ADRCs is a study limitation and would have provided additional information whether the ADRCs reached the AAA wall. As rats were only terminated at the end of the experiment, we are uncertain whether it would have been possible to detect ADRCs in the wall at this timepoint. Lines 254-257 were added to reflect this comment.
It is also worth mentioning that most studies including ours, design their experiments using a single one-time injection. Although our study did not find a positive association between a single injection and attenuation of AAA, other studies have shown that a single injection was able to attenuate AAA development (11).
- The title needs to be changed to “A single injection of ADRCs do not prevent AAA formation in rats in a randomized-blinded design”.
RESPONSE: Agree, thank you for this suggestion. The title has been changed as suggested.
Reviewer 2 Report
Comments and Suggestions for Authors
This article investigates the impact of adipose tissue-derived regenerative cells on abdominal aortic aneurysms in rats. The conclusion is that a single infusion of one million stem cells did not show significant differences in the morphology, tissue structure, and inflammatory cell infiltration of the aneurysm compared to the non-injected control group when evaluated on day 28. The preclinical test conclusion suggests that adipose tissue-derived regenerative cells do not have a preventive effect on abdominal aortic aneurysms.
Major issues are raised:
1. There is no qualitative analysis to characteristic the regenerative cells derived from rat adipose tissue.
2. The study does not compare different injection cell quantities, injection frequencies, injection route (arterial vs venous) and the fate of the cells post-injection, limiting the depth of the article.
3. From a disease prevention perspective, there is no further analysis of related protein or gene expression.
4. In terms of data presentation, it would be clearer if there were comparative data on the morphology and size of normal rat blood vessels.
5. The appearance of the aneurysms might also be of interest to readers.
Author Response
Major issues are raised:
We thank the reviewer for the comments, these are our responses.
Reviewer 2:
- There is no qualitative analysis to characteristic the regenerative cells derived from rat adipose tissue.
RESPONSE: We have previously isolated and cultured ADRCs from rats using a slightly different isolation protocol but based on similar principles described by Quade et al. (PMID: 35163613). Although the method varied slightly, they were both based on collagen digestion. The cultured ADRCs from rats were relatively homogeneous, characterized as CD45−/CD31−/CD34−/CD44+/CD90+ cells, and exhibited adipogenic, chondrogenic, and osteogenic differentiation potential, as well as the ability to form capillary-like structures upon stimulation.
This is included in the methods section:” Lines 348- 350 changed: “These cells have previously been cultured using a slightly different isolation method and shown to be relatively homogeneous CD45−/CD31−/CD34−/CD44+/CD90+ cells with the ability to differentiate into various cell types (PMID: 35163613).”
Another article with a protocol like ours for ADSCs (PMID: 25722428) reported that about 80% of the CD11b+ cells were macrophages, indicated by the F4/80 marker. A significant majority of these cells expressed the M2 macrophage phenotype marker CD301, suggesting their role in tissue repair and anti-inflammatory responses.
Due to limited time for this revision, we were unable to perform further characterization of the cells used.
- The study does not compare different injection cell quantities, injection frequencies, injection route (arterial vs venous), and the fate of the cells post-injection, limiting the depth of the article.
RESPONSE: Yes, we agree, and we also discussed in the discussion it is a limitation of the article. However, the number of cells was based on our group’s previous successful experience locally into the corpus cavernosum in the study by Quade et al (PMID: 35163613) and in mice subcutaneously studying lymphedema in Bucan et al (PMID: 32520635) and in a rat study using the same PPE AAA model showing a significant decrease in AAA expansion 14 days after induction when 1x10^6 isolated human umbilical cord mesenchymal stem cells provided IV at post-surgery day 1 (PMID: 32486904). Suggesting timing of ADRCs delivery and cell would be effective in our experiment as well. As discussed in the discussion timing of cell delivery could also have been tested weekly as shown by Fu et al. (PMID: 23875706) using the AngII-induced AAA in apoe-/- mice where mesenchymal stem cells were applied weekly. These experiments were performed in mice and used a different model, that may explain the differences in our findings.
A single injection of stem cells was previously reported as a successful treatment option in other studies (PMID: 21908146, PMID: 27436185) as mentioned in our discussion, thus providing argumentation for a single injection model. On the other hand, it could be argued that the multiple-injection model is reasonable and more realistic for translation for human studies depending on the frequency of intervention. Arterial injections are not used in clinical routine practice or experimental settings due to high-risk embolization that could result in amputation or premature death, while venous injections are safe. However, as AAA is a chronic disease it would be beneficial if a single injection of RDRC could be used, rather than multiple injections.
The lack of cell tracking is a significant limitation of our study. However, multiple studies describe a single venous injection through the tail vein as the preferred route of administration, providing an established drug administration method. A review study by Li et al summarized reports of stem cell treatment for AAA and concluded that of the 28 studies reviewed, 21 studies in vivo administered the cells intravenously (PMID: 35209940). We discussed this more thoroughly in the discussion, on possible explanations for negative results:
Lines 251-263 ‘’Future studies would focus on multiple injection models. Arterial injections are avoided due to the high risk of embolization, leading to severe complications. Venous injections are the standard practice and are considered safe for patients in both current and future treatments. We acknowledge that not examining ADRCs' homing is a study limitation, as it would have shown whether ADRCs reached the AAA wall. Since the rats were only euthanized at the experiment's end, it's uncertain if ADRCs could be detected in the wall at that time. Additionally, the timing and duration of treatment are crucial elements. A rat study using the same PPE AAA model demonstrated a significant decrease in AAA expansion 14 days post-induction with 1x10^6 human umbilical cord mesenchymal stem cells administered intravenously on post-surgery day 1 (17). This implies that the timing and quantity of ADRCs delivery were expected to be effective in our experiment. Thus, the observed difference is likely ascribed to different subpopulation of regenerative cells used. Furthermore, weekly cell delivery, as shown by Fu et al. (21) in an AngII-induced AAA model in apoe-/- mice, could have been considered’’.
- From a disease prevention perspective, there is no further analysis of related protein or gene expression.
RESPONSE: The decision not to continue further protein or gene expression analyses was made based on the negative primary outcome (AAA growth) between controls and stem cell treated rats. Furthermore, we did not detect any significant differences in Miller elastin content nor severity score of elastin breakage in the aneurysm wall between groups. These findings suggest that no major differences in protease activity like MMP2 or MMP9 between ADRC treated rats and control rats are present., further testing was deemed unnecessary.
- In terms of data presentation, it would be clearer if there were comparative data on the morphology and size of normal rat blood vessels. Available
RESPONSE: Yes, we could have included the morphology of a normal-sized aorta. However, we did not sample non-aneurysmal abdominal aorta from our rats. We did however measure abdominal aorta by ultrasound and have reported that not affected area in the aorta, the suprarenal aorta did not increase in size, figure 1.
- The appearance of the aneurysms might also be of interest to readers.
RESPONSE: We did not collect macrographs of the aneurysms showing the shape of the aneurysm at time of termination, as no camera was attached to the dissection microscope. We prioritized to process the tissue fast once taken out of the animal to secure as close to in vivo settings as possible. We believe the ultrasound measurements of the inner luminal diameter to identify AAA growth to be the best way to determine AAA growth in this model, figure 1 is updated to reflect this comment.
Round 2
Reviewer 1 Report
Comments and Suggestions for Authors
The manuscript has been improved. No further comments.
Author Response
We thank the reviewer 1 for the previous comments. The manuscript is improved.
Reviewer 2 Report
Comments and Suggestions for Authors
It is a pity that more experimental results cannot be added to confirm the conclusion of this study, such as increasing the injection of different cell numbers. In addition, in Figure 1, the labeling suggestions for suprarenal control and ADRC group can be more clearly distinguished.
Author Response
We thank the reviewer 2 for the previous comments, figure 1 has been adjusted accordingly. It is more clearly visible.